

**Responses of soil physico-chemical properties to**
**combustion: a space for time substitution study to infer**
**how changes in climate are likely to affect response of**
**topsoil to fires**
**Samuel N. Araya[1], S. Mercer Meding[2] and Asmeret Asefaw Berhe[1,3]**
[1]Environmental Systems Graduate Group, University of California, Merced
[2]Soil, Water, and Environmental Science Dept., University of Arizona, Tucson
[3]Life and Environmental Sciences Unit, University of California, Merced
Corresponding author: S. N. Araya (e-mail: saraya@ucmerced.edu)
**Abstract**
Fire is a common ecosystem perturbation that affects many soil properties. As global fire
regimes continue to change with climate change, we investigated the effect of fire heating
temperatures on the physical and chemical properties of soils across a climosequence transect
along the Western slope of the Sierra Nevada that spans from 210 to 2865 m.a.s.l. All the soils
we studied were formed on a granitic parent material and have significant differences in soil
organic matter (SOM) concentration and mineralogy owing to the effects of climate on soil
development. The dominant vegetation from lowest to highest elevation across the transect
range from oak woodland, oak/mixed-conifer forest, mixed-conifer forest and subalpine mixed-
conifer forest. Topsoils (0 - 5 cm depth) from the Sierra Nevada climosequence were heated in
a muffle furnace at six set temperatures that cover the range of major fire intensity classes (150,
250, 350, 450, 550 and 650 °C). We determined the effects of fire heating temperature on soil
aggregate strength, aggregate size distribution, specific surface area (SSA), mineralogy, pH,
cation exchange capacity (CEC), and carbon (C) and nitrogen (N) concentrations. With increase
of temperature, we found significant reduction of total C, N and CEC. Aggregate strength also
decreased with further implications for loss of C protected inside aggregates. Soil pH and SSA



increased with increase in temperature. Most of the statistically significant changes ($p < 0.05$)
occurred at temperature ranges of 350 to 450 °C. We observed relatively smaller changes at
typical temperature ranges of prescribed fires (i.e. less than 250 °C). This study identifies
critical combustion temperature thresholds for significant physico-chemical changes in soils
that developed under different climate regimes, allowing inferences for how soils are likely to
respond to different fire intensities under anticipated climate change scenarios.
**Keywords**
Chemical properties, Climate change, Climosequence, Fire intensity, Physical properties, Soil
heating
**1    Introduction**
Fire is a common, widespread phenomenon in many ecosystems around the world (Bowman et
al., 2009). Vegetation fires burn an estimated 300 to 400 million hectares of land globally every
year (FAO, 2005). In the US alone, over 80,000 fires were reported in 2014–including about
63,000 wild-land fires, and 17,000 prescribed burns that burned over 1.5 million and 970,000
ha of land, respectively (National Interagency Fire Center, 2015). Climate and climatic
variations exert strong control on distribution, frequency, and severity of fires (Harrison et al.,
2010). Significant changes on global fire regimes are anticipated because of climate change
including an increase in frequency of fires in the coming decades (Westerling et al.,
2006a;Pechony and Shindell, 2010). However, as of yet, we only have limited understanding of
how changes in fire regimes and climate change are likely to interact to influence response to
topsoils in fire affected ecosystems to wild- or prescribed-fires.
Even though humans are responsible for causing a substantial proportion of vegetation fires
(Caldararo, 2002), vegetation fires are also natural phenomena with an important role in
maintaining the health of many ecosystems around the world. Many ecosystems in the US and
across the world depend on fires to maintain ecosystem health and productivity. In the Sierra
Nevada, vegetation fires have a major influence on the landscapes (McKelvey et al., 1996).
Lightning fires were historically common in the dry season in the upland forests of the Sierra



Nevada Mountains and also fires were used by some of the Native American tribes to modify
the environment for their needs (Parsons and van Wagtendonk, 1996).
In addition to alteration of the overland vegetation, fire also significantly affects the physical,
chemical and biological properties of soils. The degree of alteration caused by fires depends on
fire intensity and duration, which in turn depend on factors such as amount and type of fuels,
properties of above ground biomass, air temperature and humidity, wind, topography, and soil
properties of moisture content, texture and SOM content (DeBano et al., 1998). The first-order
effects of fire on soil are caused by the input of heat causing extreme soil temperatures in topsoil
(Badía and Martí, 2003;Neary et al., 1999) resulting in loss and transformation of SOM, changes
in soil hydrophobicity, changes in soil aggregation, loss of soil mass, and addition of charred
material and other combustion products (Albalasmeh et al., 2013;Rein et al., 2008;Mataix-
Solera et al., 2011). Soil temperature thresholds for some important soil transformations are
illustrated in Figure 1.
Fires also impact soil by altering and removing above-ground vegetation and topsoil biomass,
and increasing soil erodibility (Carroll et al., 2007;DeBano, 1991), subsequently leading to a
shift in plant and microbial populations (Janzen and Tobin-Janzen, 2008). Fires with longer
durations are typically expected to have more impact on soil physical and chemical properties,
and loss of SOM than fast-moving, high temperature fires (González-Pérez et al., 2004).
The aim of this study is to investigate effects of combustion temperatures on important soil
properties. Here we aim to determine how the same input of energy from fires affects topsoils
that vary significantly based on carbon content, mineralogy, and associated soil physical and
chemical properties respond to combustion. The inferences derived from this work are essential
for determining how changing climate regimes (and associated changes in vegetation dynamics
and soil properties) are likely to influence the response of topsoil to wild- and prescribed-fires.
We use a laboratory heating experiment on soils from a well-characterized climosequence in
the western Sierra Nevada mountain range in our study  to determine: (1) magnitudes of change
in soil physico-chemical properties associated with different fire heating temperatures; (2)
identify critical thresholds for major changes in soil-physico-chemical properties for soils that
significantly vary based on organic matter properties, texture, mineralogy, and other properties.;
and (3) infer the implications of changing climate on topsoil physico-chemical properties that



might experience changing fire regime. This study aims to contribute to the systematic
evaluation and development of ability to predict the effect of different intensity fires on soil
properties under changing climate scenarios.
**2   Materials and methods**
**2.1   Study site and soil description**
The Sierra Nevada ecosystems are fire-adapted systems. Fires are common perturbations that
maintain ecosystem health and plant productivity in the region. However, over the last couple
of decades, the frequency and severity of fires has been increasing due to changes in climate
(Westerling et al., 2006b). For this study, we collected soils from five sites across elevation
transect along the western slope of central Sierra Nevada (Figure 2); the sites were previously
characterized by Dahlgren et al. (1997). We selected four forested sites that are likely to
experience forest fires and a fifth lower elevation grassland site for comparison.
All the sites have a Mediterranean climate characterized by warm to hot dry summers and cool
to cold wet winters. Mean annual air temperature ranges from 16.7 °C at the lowest site located
at 210 m to 3.9 °C at the highest elevation site which is at an elevation of 2865 m. Annual
precipitation ranges from 33 cm at the lowest site to 127 cm at the highest site (Dahlgren et al.,
1997;Rasmussen et al., 2007) (Table 1).
Soils from the lowest elevation site, Vista soils (210 m), fall within the oak woodland zone
(elevations < 1008 m). This is the only soil in our study that does not have an O-horizon, the
soil has dense annual grass, however, and the A-horizon SOM originates mainly from root
turnover. Musick soils (1384 m) lie within oak/mixed-conifer forest (1008—1580 m) and
mixed-conifer forest (1580—2626 m). These soils receive the highest biomass and litter fall.
Shaver and Sirretta soils (2317m) fall within the mixed-conifer forest range zone while Chiquito
soils (2865 m) lies within the subalpine mixed-conifer forest range (2626—3200 m). These
soils have lower biomass and litter fall compared to the lower elevation soils. (van Wagtendonk
and Fites-Kaufman, 2006).
The western slope of central Sierra Nevada presents a remarkable climosequence of soils that
developed under similar parent material and are located in landscapes of similar age, relief,
slope and aspect (Trumbore et al., 1996) formed on a granitic parent material with significant



developmental differences attributed to climate. The soils at mid-elevation range (1000 to 2000
m) tend to be highly weathered while soils at high and low elevations are relatively less
developed (Jenny et al., 1949;Huntington, 1954;Harradine and Jenny, 1958;Dahlgren et al.,
1997). Among the most important changes in soil properties along the climosequence include
changes in soil organic carbon (SOC) concentration, base saturation, and mineral desilication
and hydroxyl-Al interlayering of 2:1 layer silicates. Soil pH and the concentrations of clay and
secondary iron oxides show a step change at the elevation of present-day average effective
winter snowline, i.e. 1600 m elevation (Tables 1 and 2) (California Department of Water
Resources, 1952-1962;Dahlgren et al., 1997).
**2.2  Experimental design and sample collection**
To investigate the effect of combustion temperature on physico-chemical properties of soils
with significantly different carbon contents, mineralogy, and overall development, we collected
top soils (0 to 5 cm depth) from five sites. Triplicate samples, approximately 10 m apart, were
collected from each site. The soils were air-dried at room temperature and passed through 2 mm
sieve. Prior to furnace heating, the soils were oven dried at 60 °C overnight. Soil bulk density
and field soil moisture were determined from separate undisturbed core samples collected from
each site (Table 2).
Sub-samples from each soil were heated in muffle furnace to one of six selected maximum
temperatures (150, 250, 350, 450, 550 and 650 °C). To ensure uniform soil heating and reduce
formation of heating gradient inside, the soils were packed 1 cm high in a 7 cm diameter
porcelain flat capsule crucibles. Furnace temperature was ramped a rate of 3 °C min$^{-1}$ and soils
were exposed to the maximum temperature for 30 minutes. Once cooled to touch, soils were
stored in in air-tight polyethylene bags prior to analysis.
The six heating temperatures were selected to correspond with fire intensity categories that are
based on maximum surface temperature (Janzen and Tobin-Janzen, 2008;DeBano et al.,
1977;Neary et al., 1999), that is, low intensity (150 and 250 °C), medium intensity (350 and
450 °C), and high intensity (550 and 650 °C). These fire intensity classes generally correspond
with thresholds for important thermal reactions in soils observed by differential thermal
analyses (Giovannini et al., 1988;Varela et al., 2010;Soto et al., 1991) The relatively slow



heating rate of 3 °C min$^{-1}$ is recommended for laboratory based fire simulation experiments
(Giovannini et al., 1988;Terefe et al., 2008;Varela et al., 2010) to prevent sudden combustion
when soil's ignition temperature is reached at about 220 °C (Fernández et al., 1997, 2001;Varela
et al., 2010). The identical treatment of all soils, then allows for comparison of how the soils
from the different climate regimes are likely to respond to fires. Furthermore, once the set
temperature is reached, samples were exposed to that temperatures for a period of 30 minutes.
This is approximately equivalent to the time it takes to burn off small dry logs (Stoof et al.,
2010;Chandler et al., 1983) and 30 to 40 minutes has become the standard in laboratory soil
heating experiments (for example Varela et al., 2010;Giovannini, 1994;Fernández et al.,
2001;Zavala et al., 2010).

## 11  2.3   Laboratory analysis

Soil color was measured using the Munsell Color Charts. Dry color was measured from air-
dried samples and moisture was added to same sample for moist color measurement. Dry-
aggregate size distribution was measured by sieving. Samples were dry sieved into three
aggregate size classes: 2–0.25 mm (macro-aggregates), 0.25–0.053 mm (micro-aggregates) and
<0.053 mm (silt and clay sized particles). These aggregate size classes were selected to enable
comparison with other studies that investigated the effect of different natural and anthropogenic
properties on soil aggregate dynamics and aggregate protected organic matter (Six et al 2000).
Water-stable aggregate percent was measured by wet-sieving methods of Nimmo and Perkins
(2002) using a wet-sieving apparatus (Eijkelkamp Agrisearch Equipment, Giesbeek, The
Netherlands) with 0.25mm mesh size. Four grams of soil is weighed into sieve and slowly pre-
wetted by capillary rise. Sample was then wet-sieved with an up-down motion with a vertical
distance of 1.3 cm and a rate of 35 cycles per minute. Soil passing through sieves was collected
in the cans and weighed after evaporating the supernatant water in oven ($M_1$). The samples
remaining in the sieve were then subjected to a second round of wet-sieving using another set
of cans filled with dispersing solution (2 gL$^{-1}$ of sodium hexametaphosphate for the soils with
pH >7 and 2 gL$^{-1}$ NaOH for the soils with pH <7). Samples were wet-sieved until all particles
smaller than the screen opening pass. Mass of soil collected in the second set of cans ($M_2$) was
determined by evaporating supernatant solution in oven and subtracting the weight of the
dispersing-agent. The water-stable aggregate fraction was calculated as:



$WSA = \frac{M_2}{M_1 + M_2} \times 100\%.$
Specific surface area was measured using automated $N_2$-BET analyzer (Micromeritics Tri-Star
3000, Micromeritics Instrument Corporation, Norcross, GA, USA). For this procedure,
approximately 1 g of soil samples were oven dried at 60 °C for 36 and out-gassed for another
30 minutes using a flow of $N_2$ gas with outgassing station mantle set to a temperature of 105
°C. Measurement was done using ultra-high purity $N_2$ gas and the instrument was set to seven
point measurement. The isotherm is analyzed using Micromeritics software.
Soil mineralogy was measured from X-ray diffraction analysis (XRD) using PANalytical Xpert
Pro diffractometer (PANalytical Inc., Westborough, MA, USA). We used the PANalytical
Xpert Pro software for identification of mineral phases and Rietveld refinement for
quantification (Schulze and Dixon, 2002;Rietveld, 1969). Soil samples were ground to fine
powder consistency using ball-mill (8000M MiXer/Mill, with 65 ml stainless steel grinding vial
set, SPEX SamplePrep, LLC, Metuchen, NJ, USA) and oven dried at 60 °C for over 36 hours.
Samples were scanned at generator setting of 45 mA by 40 kV. Scan start position was set to 5°
2θ and end position was set to 120 2θ. Scan step time was set to 10 seconds at step interval size
of 0.0170° 2θ. Two or three replicate measurements were run for each sample and samples were
measured in random order.
Soil pH was measured 1:2 solid:solution ratio mixtures in a deionized water and 0.01 M $CaCl_2$
solution. Five grams of soil was mixed by shaking with 10 ml of solution and allowed to stand
for 30 minutes with stirring every 10 minutes. The pH reading was taken by placing electrodes
directly in the sediment slurry immediately after stirring (Thomas, 1996).
Cation exchange capacity (CEC) was measured by the barium exchange method. Barium was
used to quantitatively displace soil exchangeable cations, and excess barium was removed by
four deionized water rinses. A known quantity of calcium is then exchanged for barium and
excess solution calcium is measured in order to determine CEC by the difference in the quantity
of the calcium added and the amount left in the resulting solution. The method has a detection
limit of 2.0 $cmol_c$/kg (Rible and Quick, 1960).
Elemental concentrations of carbon (C) and nitrogen (N) were measured using an elemental
combustion system (Costech ECS 4010 CHNSO Analyzer, Costech Analytical Technologies,





Valencia, CA, USA) that is interfaced with a mass spectrometer (DELTA V Plus Isotope Ratio
Mass Spectrometer, Thermo Fisher Scientific, Inc, Waltham, MA, USA). For the analyses, air-
dried < 2 mm soil samples were ground to powder consistency on a ball-mill (8000M
MiXer/Mill, with a 55 ml tungsten Carbide Vial, SPEX SamplePrep, LLC, Metuchen, NJ, USA)
and oven dried at 60 °C for over 36 hours. The values for C and N concentration were corrected
for oven dried weights by oven-drying subsamples at 105 °C.
*Statistical Analysis*
All quantitative results are expressed as means of three replicates ± standard error, unless
otherwise indicated. Differences of means were tested by Analysis of Variance (ANOVA) and
pairwise comparison of treatments done using Tukey's HSD test at $p < 0.05$ significance level.
The ordinary linear regression technique was used to examine relationships between soil
properties. All statistical analysis were performed using R statistical software (r-project.org).
**3   Results**
**3.1   Soil color**
We observed a marked soil color change, as inferred using the Munsell color system, with
increasing heating temperature (Figure 3). With increase in heating temperature, all the soils
exhibited a similar trend in color change. As the heating temperatures increased, the soils
initially got darker with, reaching their darkest color in mid temperatures (250 - 350 °C when
dry and 250 - 450 °C when moist). At higher temperatures, the soils became markedly lighter
and became increasingly reddish in color (with hue changing from 10YR to 7.5YR at
temperatures above 550 °C). Color change patterns were similar for dry and for moist soils
except for the marked color change occurring at 450 °C in dry soils and at 550 °C in moist soils.
Across the heating temperature range, Vista (210m) soils showed the least pronounced increase
in darkness at 350°C while Shaver (1737m) soils showed the most pronounced darkening at that
temperature range (from dry color of 10YR 5/2 unburned to 10YR 3/3 by 350°C). At higher
temperatures, Musick soils (1384m) showed the largest change in dry soil color going from
10YR 2/2 at 350 °C to 7.5YR 6/6 at 650 °C.





## 3.2 Mass loss

Mass loss was proportional to heating temperature in all the soils. For the high and low elevation soils, statistically significant mass loss, compared to unburned soils, was observed above 350ºC. In contrast, significant mass loss was observed for the two mid elevation soils of Musick (1384m) and Shaver (1737m) starting 250ºC. There was no significant mass loss at temperatures above 450ºC for all soils. For all our soils, the steepest mass loss was observed between temperatures of 250 and 450 °C (Figure 4). Vista (210m) soils showed the lowest mass loss with heating while Musick soils (1384 m) showed the highest mass loss with heating.

## 3.3 Aggregate stability and size distribution

Aggregate stability generally decreased with temperature for all soils. While aggregate stability seemed to decrease in an almost uniform manner with increase in temperature for the lower to mid elevation soils, the higher elevation Sirretta (2317m) and Chiquito (2865m) soils showed a stepwise decrease in aggregate stability at 250 °C and 350 °C respectively. At higher temperature heating aggregate stability for the two soils showed only a small decrease from these two temperatures. Statistically significant decrease in aggregate strength, compared to unburned samples, was observed only at higher temperatures above 350 °C for Sirretta (2317m) and Chiquito (2864m) soils, and above 450 °C for Musick (1384m), Vista (210m) and Shaver (1737m) soils (Figure 4).

Although not statistically significant, all soils showed a decrease in macro-aggregate fraction accompanied by increase in micro-aggregate and silt-clay sized fractions (Figure 5). For the two lower elevation soils (Vista and Musick) the decrease in macro-aggregate fraction was over 10% and less than 5% for all the other soils. Only Musick (1384m) soils showed a statistically significant decrease in macro-aggregate fraction between 150 and 350 °C temperatures.

## 3.4 Specific surface area

For all soils, we observed a stepwise increase in specific surface area (SSA) for samples were heated to between 250 to 450 °C (Figure 4). Changes in SSA between soils heated below 250 °C and those heated above 450 °C were statistically significant at $p<0.05$ for all soils, except the high elevation soils Sirretta (2317 m) and Chiquito (2865 m). Sirretta soils showed a lot of



variability and did not show any significant change in SSA throughout the temperature range
while the Chiquito soil showed statistically significant increase between low temperature 150 –
250 °C and higher temperature 350 – 550 °C range. The pattern of change in SSA with
temperature was similar for all soils. The lowest SSA was recorded for all soils when soils were
heated at 250 °C, and  highest SSA was observed at 350 °C (for Musick and Chiquito soils) or
450 °C (Vista and Shaver soils).
**3.5   Soil mineralogy**
The bulk soil XRD results of changes in soil mineralogy in response to heating are presented
for basic mineral groups as: feldspar (microcline and orthoclase); plagioclase (albite and
oligoclase); amphibole; mica/illite (biotite); kaolinite; gibbsite; and expandable phyllosilicate
(montmorillonite and vermiculite). We identified vermiculite with low confidence, since we did
not correct with oriented clay treatments, hence it is not certain if the identified peaks are indeed
representative of vermiculite, chlorite, or both. The XRD diagrams showed some significant
transformations in soil mineralogy with heating, with shifted peaks at higher temperatures
suggesting transformation of clay minerals.  Layer silicates appeared to collapse structurally,
possibly due to dehydration and the removal –OH (Figure 6). Summary mineral composition
changes identified from XRD analysis using Rietveld method are presented in Figure 7. Across
all the soils, the largest mineral composition change with increased heating temperatures were
observed for kaolinite that experiences loss at temperatures above 550ºC. Gibbsite was not
found in soils that it was originally present after 450ºC. Furthermore, mica/illite, plagioclases
and amphibole mineral groups changed consistently with increasing temperature. The largest
change in soil mineralogy with heating was observed for the mid-elevation Music (1384m) and
Shaver (1737m) soils that are also among the most developed soils with highest proportions of
the 1:1 clay minerals (kaolinite).
**3.6   Soil pH**
With increase in temperature, all soils showed a similar pattern of increase in pH (Figure 8).
For all soils the largest increase in pH (2.5 – 5 units) occurred between 250 and 450 °C. All the
soils started out with slightly to moderately acidic pH and with the exception of Chiquito (2865)
soils all soils became alkaline at temperatures above 450 °C. The largest increase in pH was



observed for the Musick (1384 m) soils which reached a pH of 10 at temperatures above 550
°C.
**3.7  Cation exchange capacity**
The CEC of our soils ranged from an average of 6 cmol$_c$/kg for Chiquito (2865 m) soils to 25
cmol$_c$/kg for Musick (1384 m) soils. With increase of heating temperature all soils showed
continued decrease of CEC. With the exception of Musick (1384m) soils, CEC eventually
dropped to below our detection limit (2 cmol$_c$/kg) at temperature above 550 °C (Figure 8). For
the poorly weathered Chiquito (2865 m) soils, CEC was below 2 cmol$_c$/kg at temperatures 250
°C and above. For the rest of the soils, statistically significant changes in CEC ($p<0.05$) occurred
at 450 °C with the exception of Musick (1384 m) soils which showed statistically significant
drop at 250 °C and again 350 °C. At 350 °C, all the soils except Musick (1384 m) showed a
slightly higher CEC than at 250 °C thus interrupting continuous pattern of CEC decrease with
increase in temperature.
**3.8  Carbon and nitrogen concentration**
The initial concentration of C range from less than 2% (for the Vista soil, 210 m) to over 7 %
(for the Musick soils, 1384 m). Soil C concentration decreased with increase in temperature
(Figure 9) with the largest decrease occurring between temperatures of 250 and 450 °C. At 450
°C, all soils lost more than 95% of their initial C and changes at higher temperatures were small
and statistically insignificant ($p<0.05$). Soil's C:N ratio ranged from 10 (Vista soils, 210 m) to
29 (Musick soils, 1384 m). For all soils C:N ratio decreased with increase in heating temperature
in a similar pattern as what we observed for the changes in C concentration (Figure 9).
The loss of C and N from soils due to heating showed a similar response among all five soils.
After 250 °C, all the soils lost more than 25% of their initial C (except Shaver soils that lost
only about 10%). At 350 °C all soils lost 50 to 70% of C. Combustion at 450 °C led to loss of
more than 95% of their initial C for all soils in this study. Loss of N was lower than that of C.
At temperatures greater than 550 °C there was 5 to 15% of soil N still remaining. Consequently,
we observed a decrease of C:N ratio with increase in heating temperature. All soils continued
to lose about 15% soil N for every 100 °C increase and maintained more than 60% of their N at



heating temperatures up to 350 ºC. After heating at 450 °C, all soils lost more than 60% of the
initial soil N and 85% by 550 °C.
**4    Discussion**
Topsoil layer is most affected by extreme temperature during vegetation fires. Our results show
significant changes in soil properties as a result of temperature exposure. Our findings
demonstrate that alterations and loss of SOM in topsoil, rather than alterations to soil minerals,
was the most important driver for the observed changes in soil physico-chemical properties after
combustion. Our XRD analysis shows notable changes in soil mineralogy only after the soils
were heated to about 450 to 550 °C (Figure 7). In upland ecosystems, such as the Sierra Nevada
Mountains, the soils typically have low clay content and low concentration of secondary
minerals (Neary et al., 1999;Ubeda and Outeiro, 2009). In addition, these upland temperate
ecosystems also tend to have relatively high concentration of SOM, including a fairly well
developed O-horizon. Consequently, strong relationships are observed between SOM
concentrations and the soils' physical and chemical properties. Simple linear regression
analyses between C concentration changes and other soil physical and chemical changes for our
study soils shows that more than 80% of the variability in mass loss, aggregate strength, SSA,
pH, CEC and N concentrations are associated with changes in C concentration at the different
heating temperatures. Table 3 summarizes the correlation coefficients of soil property changes
with changes in C concentration.
The changes in soil color observed were consistent with the charring of SOM which leads to
darkening of the brownish color of the soils. At temperatures over 450 °C, the near complete
removal of SOM by combustion and the addition of ash products likely explains the observed
lighter color of soils. The increase in Munsell chromas and reddening of soils at these high
temperatures has been noted in previous works (e.g. (Giovannini et al., 1988;Ulery and Graham,
1993;Ketterings and Bigham, 2000) and is likely a result to oxidation and transformation of
iron oxides in a manner analogous to aging of soils and transformations of mineral soil after
intense weathering.
The extent of mass loss in top soil layers due to vegetation fires is strongly correlated to SOM
combustion (Rein et al., 2008). In all of our soils, statistically significant mass loss ($p<0.05$)
occurred within the temperature ranges of SOM combustion. Proportion of soil mass loss with



temperature was proportional to initial C concentration of the soils, Musick (1384m) soils,
which had the highest initial C concentration (7%) had the steepest soil mass loss and lost 15%
of mass at 550ºC while Vista (210m) which had less than 2% C concentration showed the
smallest mass loss losing less than 5% mass at even at the highest temperature.  The influence
of other mass loss mechanisms likely account for mass loss outside of the temperature range
where SOM plays a dominant role, dehydration processes are most likely responsible for mass
loss at temperatures below 250ºC where the loss and transformation of SOM is minimal. Mass
loss at higher temperatures where most of SOM has been combusted (i.e. >450°C) is likely
dominated by charring, ashing and volatilization processes.
Most studies report significant soil aggregate stability reduction with fire heating (e.g. Zavala
et al. (2010);Arcenegui et al. (2008)), however, contrasting findings are also reported in other
studies. Mataix-Solera et al. (2011) explains that increase in aggregate strength with heating is
possible in clay-rich soils where main cementing agents are inorganic minerals such as calcium
carbonates and metallic oxides which would fuse under fire heating increasing aggregate
stability. Another possibility is, soils which initially had weak hydrophobicity may show
increased aggregate stability due to increase in hydrophobicity, in such cases however aggregate
strength would decline at higher severity fires as hydrophobicity is destroyed.
Aggregate stability in all our soils generally decreased with increase in heating temperature.
Although we did not find it to be statistically significant difference, it is worth noting, that the
lowest elevation soil (Vista, 210 m) showed a trend of aggregate stability increase up to 350 °C
while the high elevation soils, Sirretta (2317 m) and Chiquito (2865 m), showed a slight increase
in aggregate stability at 150 °C. Increase in hydrophobicity at these temperatures is the most
likely cause, substantial hydrophobicity was apparent with Chiquito soils (2865 m) heated at
250 °C where resistance to slaking was remarkably evident during aggregate stability test.
Soil specific surface area (SSA) is an important soil  property that affects soil adsorption, ion
exchange capacity, reactivity, aggregation and porosity (Feller et al., 1992). SSA of soil is
largely dictated by clay-size particles and SOM (Carter et al., 1986). The increase in SSA with
heating that we observed in this work is most likely the result of physical disintegration and
charring of SOM, especially at temperatures below 500 °C. Changes in soil mineralogy are not
likely to be responsible for the changes in SSA we observed. XRD analysis showed notable

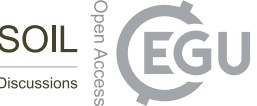

mineralogical changes only at temperatures above 450 °C where the kaolinite peak disappeared
at 550 °C and the phyllosilicate peaks were diminished or disappeared at 450 – 550 °C (e.g.
Figure 6). Because of combustion and heat-induced dehydration, larger organic matter particles
are likely to have fragmented and reduced in size with increase in heating temperature leading
to increase in bulk soil SSA. Furthermore, removal of organic matter from mineral surfaces due
to combustion may also increase surface area by reducing overall size of particles. At higher
temperatures, XRD spectra showed some collapse of mineral complexes through dehydration
and de-hydroxylation of clay minerals that may reduce mineral particle size and increase surface
area. These changes in mineralogy might have a significant effect on SOM. A study by Rosa et
al. (2013) found that soils released more organic compounds during pyrolysis when the soils
were treated with HF acid, suggesting that mineral complexes play a role in protecting organic
compounds from combustion. The collapse of mineral complexes we observed and the decrease
in aggregate strength is likely to have enhanced thermal oxidation of SOM. However
mineralogical changes would play a more important role affecting soil properties at these high
temperatures since SOM has been almost completely removed.
Soil pH generally increases with fire heating (Ubeda and Outeiro, 2009;Badía and Martí,
2003;Chandler et al., 1983). In a similar heating experiment as we followed in our study,
Fernández et al. (1997) observed a pH increase of 1.7 at 350 °C and 2.35 at 490 °C, where the
authors attributed the change in pH to denaturation of organic acids, the release of base cations
from combustion (K- and Na-hydroxides, Mg- and Ca-carbonates), the deposition of ashes and
loss of hydroxyl groups from clays (Certini, 2005;Badía and Martí, 2003). In our soils, the
higher elevation soils (Shaver, Sirretta and Chiquito) showed a statistically insignificant
decrease of 0.3 to 0.5 pH units (measured in water) at 250 °C. The change in pH in our high
elevation soils was consistent to previous results of Badía and Martí (2003);Terefe et al. (2008)
that found similar initial decrease. Terefe et al. (2008) hypothesized that this may be due to the
combined effect of desiccation and heating effect which favor proton-reducing oxidation
reactions. And the fact that this initial increase occurs below the temperature for start of
combustion of organic acids means contribution of SOM to pH increase (organic acid
denaturation and ash liming effect) was absent at this temperature. In a similar heating
experiment Badía and Martí (2003) found an increase in electric conductivity and soluble Ca
along with decrease in pH at 250 °C. Such increase in soluble cations might explain our findings



where we observed a decrease in pH when measured in water but not in CaCl$_2$ suggesting that
the decrease in pH might have to do with increase of soluble salts with heating up to 250 °C.
The capacity of soil to exchange positively charged ions between soil and soil solution (CEC)
decreased with increasing temperature. CEC of soils is a result of surface charges associated
with secondary clay minerals and SOM (Sparks, 2005), and in our study soils, Dahlgren et al.
(1997) had previously reported a strong relationship of CEC with soil organic carbon and clay
concentrations. Different authors have attributed loss of CEC during heating mainly with loss
of SOM (Ubeda and Outeiro, 2009;Fernández et al., 1997) partly because CEC loss starts to be
observed at temperatures above 200 °C with little or no decrease at lower temperatures where
SOM is not affected (Soto and Diazfierros, 1993;Nishita and Haug, 1972). The slight increase
of CEC we observed at 350 °C may be due to the steep increase of specific surface area at that
temperature (Figure 4). The additional surface for cation adsorption might have to an extent
compensated for the loss of SOM at that temperature. Furthermore, the contribution of surface
oxidation of char products has been shown to increase CEC per unit C (Liang et al., 2006)
because of the almost complete loss of C at temperatures above 450 °C and very little loss at
temperatures below 250 °C. The soils most likely had highest concentration of charred SOM at
350 °C temperature.

## 18  4.1  Importance of the 250 – 450 °C range

Based on maximum surface temperature, fires are often classified as low, medium or high
intensity. Low intensity fires reach surface temperatures of up to 250 °C, medium intensity fires
reach surface temperatures of 400 °C, and high intensity fires reach surface temperatures above
675 °C (Janzen and Tobin-Janzen, 2008). In this study, the most significant changes of soil
chemical properties occurred at the transition between low and medium severity fires, between
250 and 450 °C. Figure 10 illustrates the changes between unburned and 650 °C burned soils
and the amount of change that occurred within 250 to 450 °C heating temperature for a range
of the variables discussed above. In all cases, the change in the 250 – 450 °C range accounts for
most of the total change observed during our combustion treatments. Among the variables we
investigated in this study, we observed changes along two general lines: (1) mass loss, SSA and
pH which showed a progressive increase with heating temperature, and (2) %C, %N, C:N ratio,
CEC, and wet aggregate stability that showed a progressive decrease with combustion



temperature (Figure 10), with the most significant changes in all cases being recorded in all
soils between 250 - 450ºC.
Temperatures below 250 °C are very critical for many processes, water is lost at 95 °C and this
has a significant effect on soil heat conduction and soil biota (Janzen and Tobin-Janzen, 2008).
However, temperatures below 200 °C have very little effect on quality or quantity of SOM. This
means low intensity fires, such as typical prescribed fires, contribute little to soil C loss.
Similarly, temperatures above 500 °C do little change to SOM, which already has been lost or
transformed into a pyrogenic product. The effect on soil inorganic particles starts at high
temperature but the significance of change on minerals is not as large (Figure 1). Hence, we
found that the most important soil changes occur the 250 – 450 °C range.
Important modifications of fire conditions that still allow for comparison of responses of
different types of soils have to be adopted to conduct the type of heating experiments that we
undertook in this work. A heating rate of 3 °C min$^{-1}$ is common in laboratory soil heating
studies, often because of technical consideration and because such slow rate prevents sudden
combustion which otherwise would happen as soil's ignition temperature is reached at about
220 °C (Fernández et al., 1997, 2001;Varela et al., 2010). However, it is important to recognize
that during vegetation fires the rate of temperature increase experienced by the topmost layer
of soil that is exposed to fire can be significantly higher. The rate of heating alone might have
additional significant effects on soil properties beyond what we observe here. For example,
Albalasmeh et al. (2013) found that slow rate of heating underestimates soil aggregate
destruction of moist soils due to a slower buildup of pore-pressure.
**4.2   Climate Change Implications**
Investigation of the response of climosequence soils to different heating temperature in this
study enables us to infer how changes in climate (and associated changes in soil properties) are
likely to alter the effect of fires on topsoil physical and chemical properties. Even though the
general pattern of change in soil physical and chemical conditions with increasing combustion
temperature were mostly consistent for all soils along our study climosequence, the actual
magnitude of change in the investigated variables was not. Hence, these finds lead us to
conclude that climate change is likely to alter the response of topsoil properties to different fire



regimes. Along our study climosequence, we observed critical differences in response of
topsoils based mostly on concentration OM in soil and soil development stages of each soil --
both variables that are expected to respond to changes in climate (Berhe et al., 2012).
Consequently, changes in soil C storage associated with climate change are expected to lead to
different amounts of C loss due to fires. This is evidenced by the observed highest total mass of
C loss from the mid-elevation Musick soil that had the highest carbon stock, compared to soils
in either side of that elevation range. Anticipated changes in climate in the Sierra Nevada
mountain ranges are expected to include upward movement of the rain-snow transition line
exposing areas that now receive most of their precipitation as snow to rainfall and associated
runoff. Moving of the rain-snow transition zone higher and promotion of more intense
weathering at higher elevation zones then is likely to render more C to loss during fires. As we
found in this study, more than 80% of the variability in mass loss, aggregate strength, SSA, pH,
CEC and N concentrations is associated with changes in C concentration at the different
combustion temperatures (Table 3). Improving our understanding of how topsoil properties are
likely to respond to changes in climate becomes even more critical when we recognize that C
concentration in soil is likely to respond quickly to changes in climate, compared to other soil
physical and chemical properties (Berhe et al., 2012). Furthermore, the long-term fate of soil
carbon in fire-affected ecosystems is also likely to be accompanied by changes in microbial
community composition and OM decomposition kinetics (Holden et al., 2015;Tas et al., 2014)
which are likely to have further implications for nutrient availability post-fire (Johnson et al.,
2007b;Johnson et al., 1997).
The different responses of soil aggregation in our climosequence to the treatment temperatures
also suggest potential loss and transformation of the physically protected C pool in topsoil.
Degradation of aggregates during fire (Albalasmeh et al., 2013) is likely to render aggregate-
protected C to potential loses through oxidative decomposition, leaching and erosion.
Moreover, in systems such as the Sierra Nevada where steep slopes and organic matter-rich
topsoils dominate, movement of the rain-snow transition zone upward is likely to increase
proportion of precipitation that occurs as rain. The kinetic energy of raindrops and observed
increase in hydrophobicity of soils post-fires (Johnson et al., 2007a;Johnson et al., 2004) can
lead to higher rates of erosional redistribution of especially the free light fraction or particulate
C that is not associated with soil minerals (Stacy et al., 2015). Moreover, the important





differences in changes in pH, mineralogy, CEC in response to heating at different temperatures
that we observed for soils along the climosequence suggest that changes in temperature are
likely to lead to different effects on soil chemical properties in soils after fires.
Finally, with changes in climate it is anticipated that fires will increase in severity (Westerling
et al., 2006b). Our findings of important changes in soil physical and chemical properties
occurring between 250-450ºC are important for recognizing that critical transformations of
topsoil physical and chemical properties are likely to occur when, as a result of climate change,
systems that are adapted to low severity fires experience medium to high severity fires.
**5 Conclusion**
Findings of this study showed that changes in soil properties during heating are closely related
to changes in C concentrations in soil. The temperatures most critical to C loss and alteration
were found to be 250 °C, where charring of organic matter starts and 450 °C where most of the
SOM is combusted. Most soil properties exhibited a steep change in this temperature range.
Soil aggregate stability, CEC, and C and N concentrations significantly decreased with
increased combustion temperature while soil pH and SSA significantly increased. The most
important effect of combustion on soil mineralogy as observed by XRD analysis was the
collapse of kaolinite, which was undetectable at temperatures above 500 °C.
This study presented the effects of heat input on topsoil properties. The study is necessary to
understand the changes that occur under fires that result in heating of soil without additional
variables such as the addition of charred plant material and ash, and the influence of soil
moisture. Findings from this study will contribute towards estimating the amount and rate of
change in carbon and nitrogen loss, and other essential soil properties that can be expected from
topsoil exposure to different intensity fires under anticipated climate change scenarios.
**Acknowledgements**
The authors would like to thank Prof. Randy A. Dahlgren for providing us with geo-references
for the study sites, background data, and for his comments on an earlier version of this
manuscript. We thank Drs. Marilyn Fogel and Christina Bradley for their help and expertise in
elemental analysis of C and N; and Dr. Samuel Traina for his comments on an earlier version





of this manuscript. The research was funded by a UC Merced Graduate Research Council grant
and NSF grant (EAR-1352627) to AAB.

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

Thomas, G. W.: Soil pH and soil acidity, in: Methods of Soil Analysis. Part 3, Chemical
Methods, edited by: Sparks, D. L., Page, A. L., Helmke, P. A., and Loeppert, R. H., Soil Science
Society of America/ American Society of Agronomy, Madison, Wisconsin, 1996.
Trumbore, S., Chadwick, O., and Amundson, R.: Rapid exchange between soil carbon and
atmospheric carbon dioxide driven by temperature change, Science, 272, 393-396, 1996.
Ubeda, X., and Outeiro, L. R.: Physical and Chemical Effects of Fire on Soil, in: Fire Effects
on Soils and Restoration Strategies, edited by: Cerda, A., and Robichaud, P. R., Land
Reconstruction and Management, Science Publishers, Enfield, NH, USA, 2009.
Convert       Munsell       colors       to       computer-friendly       RGB       triplets:
http://casoilresource.lawr.ucdavis.edu/software/r-advanced-statistical-package/color-
functions/convert-munsell-colors-computer-friendly-rgb-triplets/.
Ulery, A. L., and Graham, R. C.: Forest-fire effects on soil color and texture, Soil Sci Soc Am
J, 57, 135-140, 1993.
van Wagtendonk, J. W., and Fites-Kaufman, J. A.: Sierra Nevada Bioregion, in: Fire in
California's Ecosystems, edited by: Sugihara, N. G., van Wagtendonk, J. W., Shaffer, K. E., and
Thode, A. E., University of California Press, Berkeley, CA, USA, 2006.




Varela, M. E., Benito, E., and Keizer, J. J.: Effects of wildfire and laboratory heating on soil
aggregate stability of pine forests in Galicia: the role of lithology, soil organic matter content
and water repellency, Catena, 83, 127-134, 10.1016/j.catena.2010.08.001, 2010.
Westerling, A. L., Hidalgo, H. G., Cayan, D. R., and Swetnam, T. W.: Warming and earlier
spring increase western U.S. forest wildfire activity, Science, 313, 940-943,
10.1126/science.1128834, 2006a.
Westerling, A. L., Hidalgo, H. G., Cayan, D. R., and Swetnam, T. W.: Warming and earlier
spring increase western US forest wildfire activity, Science, 313, 940-943, 2006b.
Zavala, L. M., Granged, A. J. P., Jordán, A., and Bárcenas-Moreno, G.: Effect of burning
temperature on water repellency and aggregate stability in forest soils under laboratory
conditions, Geoderma, 158, 366-374, 10.1016/j.geoderma.2010.06.004, 2010.



**Tables**
Table 1 Soil classification and site description for the five sites along elevational transect in the western slopes of
the Sierra Nevada (adapted from Dahlgren et al., 1997)

| Soil Series | Elevation (m) | Ecosystem | MAT[a] (°C) | MAP[b] (cm) | Precip[c] | Dominant vegetation (listed in order of dominance) | Soil taxonomy (family) |
|---|---|---|---|---|---|---|---|
| Vista | 210 | Oak woodland | 16.7 | 33 | Rain | Annual grasses; *Quercus douglasii*; *Quercus wislizeni* | Coarse-loamy, mixed, superactive,thermic; Typic Haploxerepts |
| Musick | 1384 | Oak/mixed-conifer forest | 11.1 | 91 | Rain | *Pinus ponderosa*; *Calocedrus decurrens*; *Quercus kelloggii*; *Chamaebatia foliolosa* | Fine-loamy, mixed, semiactive, mesic |
| Shaver | 1737 | Mixed-conifer forest | 9.1 | 101 | Snow | *Abies concolor*; *Pinus lambertiana*; *Pinus ponderosa*; *Calocedrus decurrens* | Coarse-loamy, mixed, superactive, mesic; Humic Dystroxerepts |
| Sirretta | 2317 | Mixed-conifer forest | 7.2 | 108 | Snow | *Pinus jeffreyi*; *Abies magnifica*; *Abies concolor* | Sandy-skeletal, mixed, frigid; Dystric Xerorthent |
| Chiquito[d] | 2865 | Subalpine mixed-conifer forest | 3.9 | 127 | Snow | *Pinus contorta murrayana*; *Pinus monticola*; *Lupinus* species | Sandy-skeletal, mixed; Entic Cryumbrept |

[a] Mean annual air temperature, calculated from regression equation of Harradine and Jenny (1958)
[b] Mean annual precipitation
[c] Dominant form of precipitation
[d] Tentative soil series





Table 2 Bulk density, water content, pH, C concentration, cation exchange capacity (CEC), specific surface area
(SSA) and particle size distribution for the five soils (mean ±standard error, n=3)

| Soil series and elevation (m) | Bulk density (g/cm³) | Gravimetric water content (%) | pH (CaCl₂) | Carbon (%) | CEC (cmol_c/kg) | SSA (m²/g) | Particle size distribution[a] (%) | | |
|---|---|---|---|---|---|---|---|---|---|
| | | | | | | | Sand | Silt | Clay |
| Vista (210) | 1.26 ±0.07 | 0.7 ±0.0 | 5.53 ±0.0 | 1.51 ±0.2 | 8.40 ±1.1 | 1.75 ±0.2 | 79 | 11 | 10 |
| Musick (1384) | 0.90 ±0.06 | 9.3 ±1.6 | 4.67 ±0.1 | 7.66 ±0.8 | 25.20 ±2.0 | 4.98 ±0.3 | 60 | 27 | 15 |
| Shaver (1737) | 0.98 ±0.06 | 8.3 ±1.1 | 4.85 ±0.3 | 2.84 ±0.2 | 10.67 ±2.1 | 3.08 ±0.3 | 80 | 15 | 5 |
| Sirretta (2317) | 0.61 ±0.09 | 9.9 ±2.2 | 4.54 ±0.1 | 4.74 ±0.8 | 12.23 ±2.6 | 6.63 ±0.8 | 80 | 15 | 5 |
| Chiquito (2865) | 1.17 ±0.03 | 6.1 ±1.9 | 3.96 ±0.1 | 4.10 ±0.2 | 6.03 ±1.8 | 1.00 ±0.04 | 80 | 16 | 4 |

[a] Particle size distribution of top soil profile from Dahlgren et al. (1997): Vista (0 – 14 cm),
Musick (0 – 29 cm), Shaver (0 – 4 cm), Sirretta (0 – 6 cm) and Chiquito (0 – 6 cm)



2      Table 3 Linear correlation coefficients of changes in soil properties with changes in C concentration

| Soil | Correlation coefficient ($r^2$) values | | | | | |
|---|---|---|---|---|---|---|
| | Mass loss | SSA | Aggregate Stability | pH (CaCl$_2$) | CEC | N concentration |
| Vista | 0.74 | 0.73 | 0.21 | 0.77 | 0.78 | 0.89 |
| Musick | 0.89 | 0.58 | 0.77 | 0.89 | 0.96 | 0.83 |
| Shaver | 0.82 | 0.58 | 0.68 | 0.74 | 0.78 | 0.93 |
| Sirretta | 0.60 | 0.34 | 0.47 | 0.67 | 0.87 | 0.86 |
| Chiquito | 0.82 | 0.62 | 0.78 | 0.88 | 0.44 | 0.87 |



**Figures**

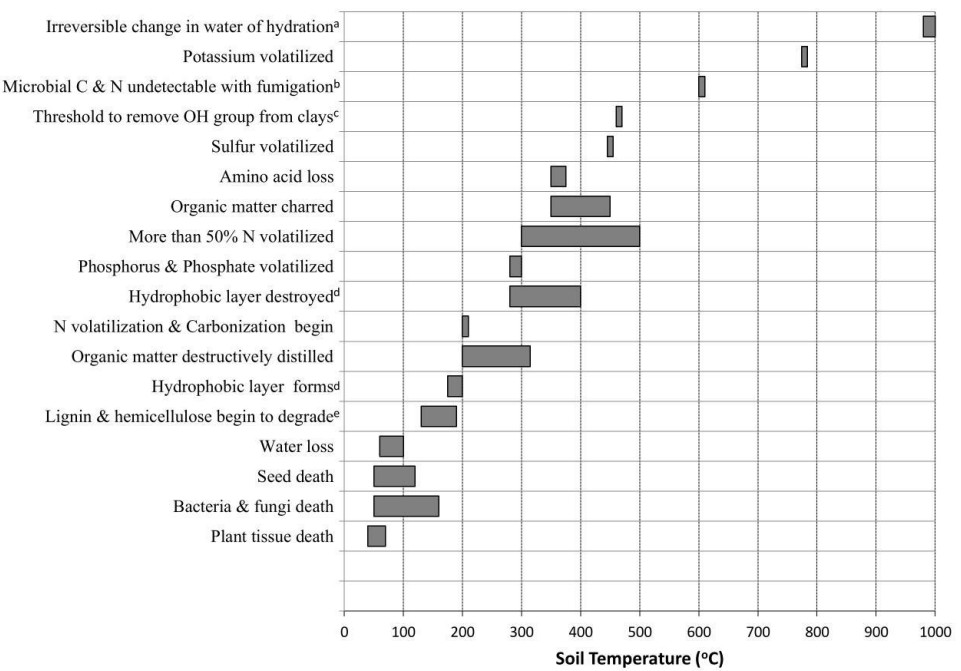

Figure 1: Temperature thresholds and ranges associated with fire heating. Figure adopted and
expanded from Massman et al. (2010). ([a] DeBano et al. (1977), [b] Diaz-Ravina et al. (1992), [c]
Giovannini et al. (1988), [d] DeBano (2000), and [e] Knicker (2007))





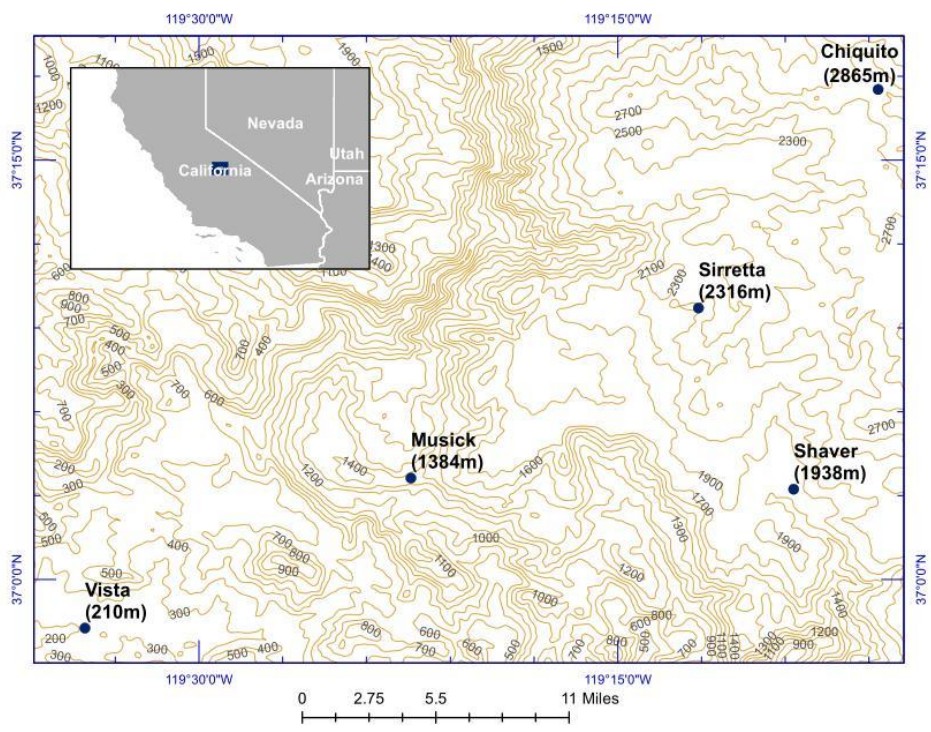

2      Figure 2: Map of the five sampling sites along elevational transect in the Western Sierra Nevada,

3      California (Base map from U.S. Geological Survey, 2015)





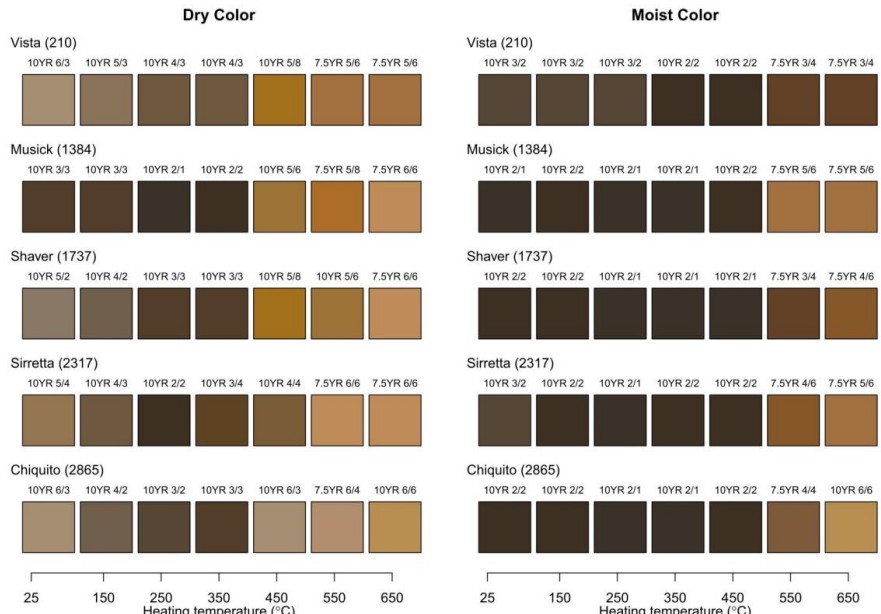

Figure 3: Soil color change across heating temperatures. Colors produced from CIExyY
colorspace equivalents to Munsell colors (Munsell Color Science Laboratory). CIExyY colors
were converted to RGB system (Rossel et al., 2006;UC Davis Soil Resource Laboratory) and
visually compared with Munsell Soil Color book for plotting.





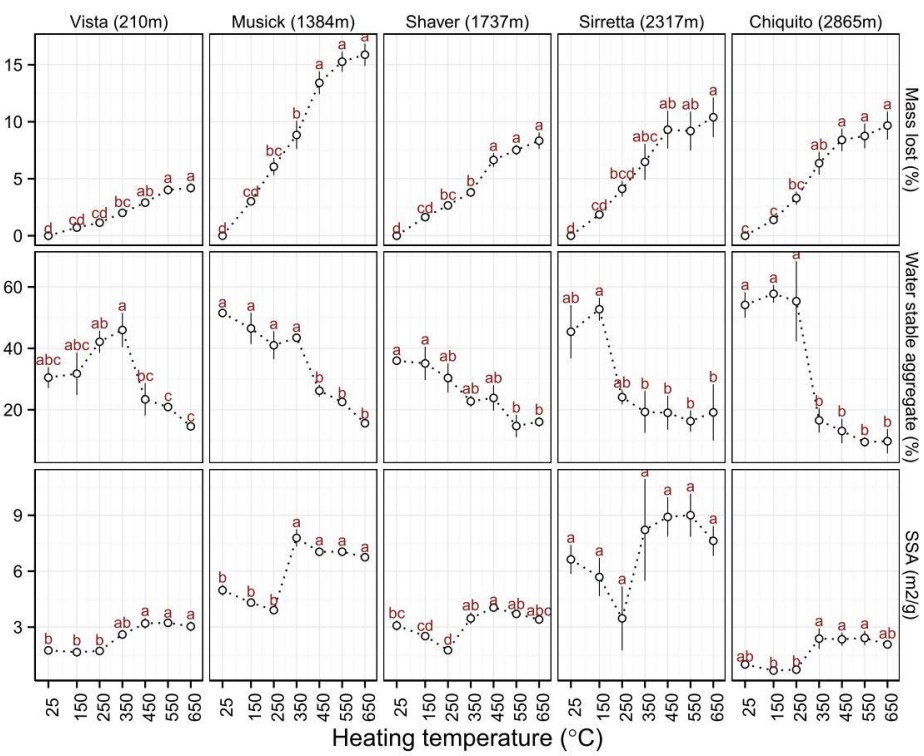

Figure 4: Percent mass lost, water-stable aggregate percent and specific surface area changes
with increase in heating temperatures. Error bars represent standard error where n=3. Different
letters represent significantly different means (p<0.05) at temperature after Tukey's HSD
testing.



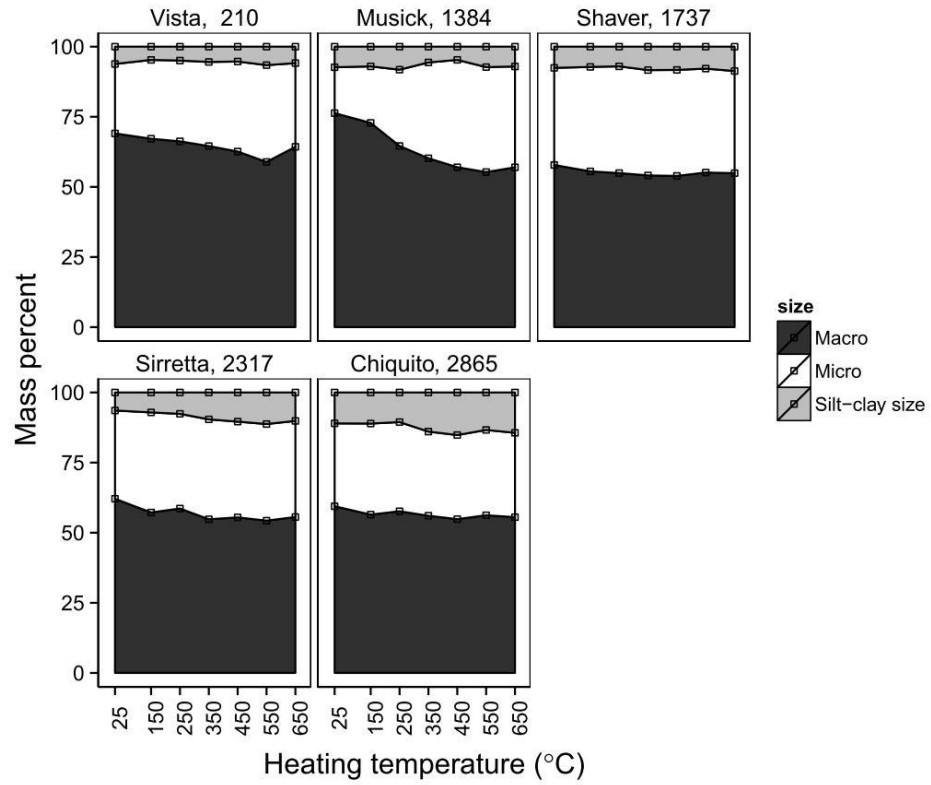

2    Figure 5: Weight fraction of aggregate sizes: macro (2-0.25 mm), micro (0.25-0.053 mm) and

3    silt-clay (<0.053 mm) sizes.



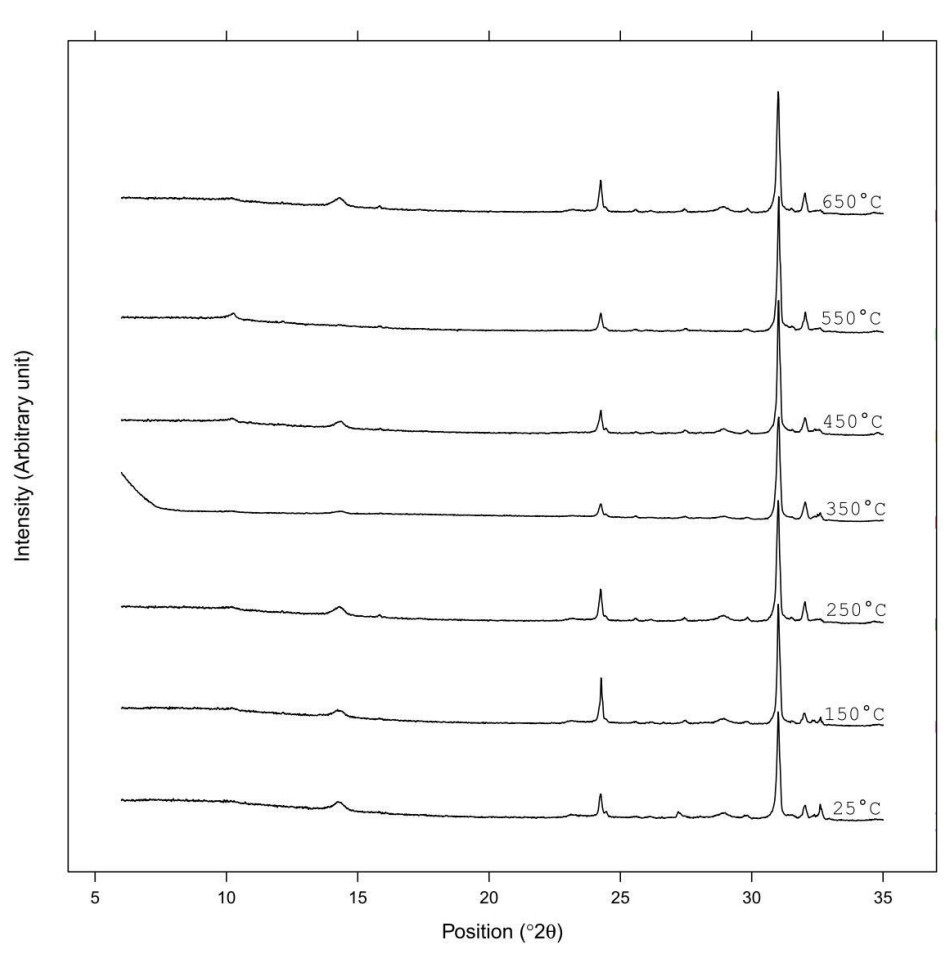

2    Figure 6: XRD diagram for Music series soils.





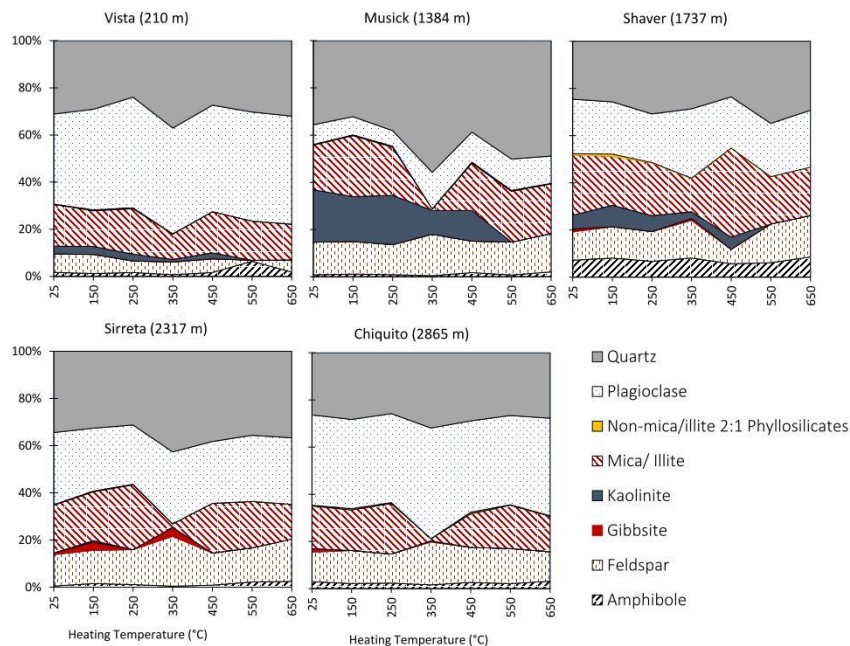

2    Figure 7: Relative amounts of minerals identified from powder XRD.

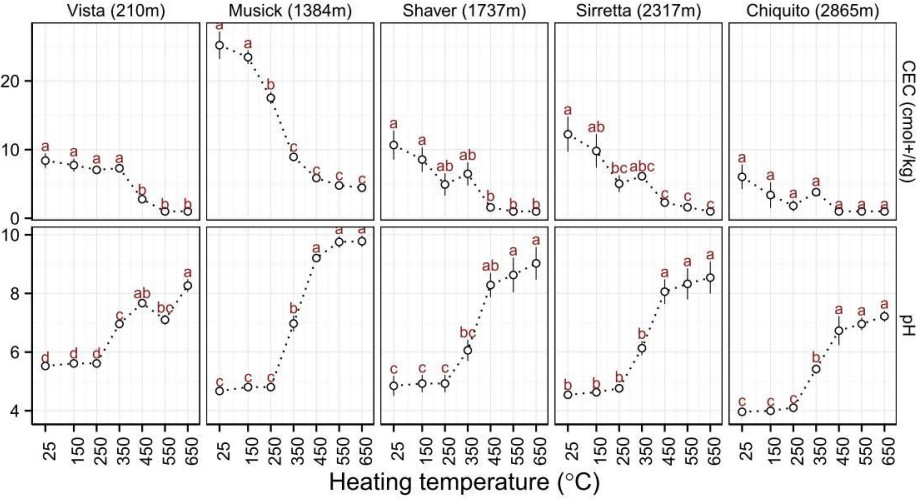



Figure 8: pH (geometric means) and cation exchange capacity (adjusted for mass loss) changes
with increase in heating temperature. CEC values below the 2 cmol$_c$/kg are assigned a value of
1 for plotting. Error bars represent standard error where n=3. Different letters represent
significantly different means (p<0.05) at each temperature after Tukey's HSD testing.

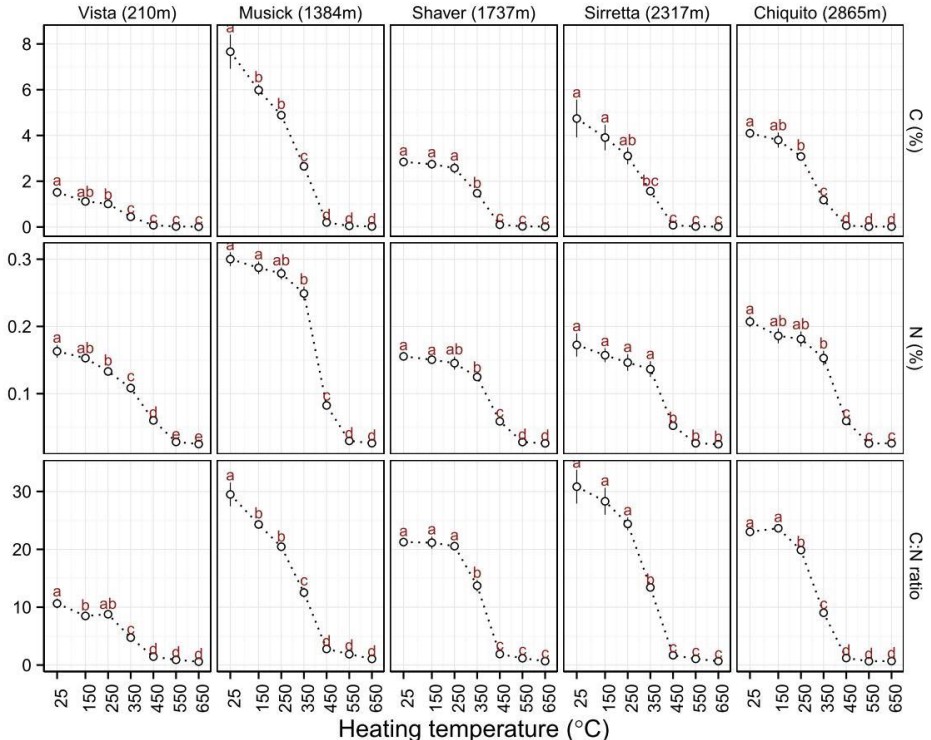

Figure 9: Carbon concentration, Nitrogen concentration and C:N atomic ratio changes with
increase in heating temperature. Error bars represent standard error where n=3. Different letters
represent significantly different means (p<0.05) at each temperature after Tukey's HSD testing.





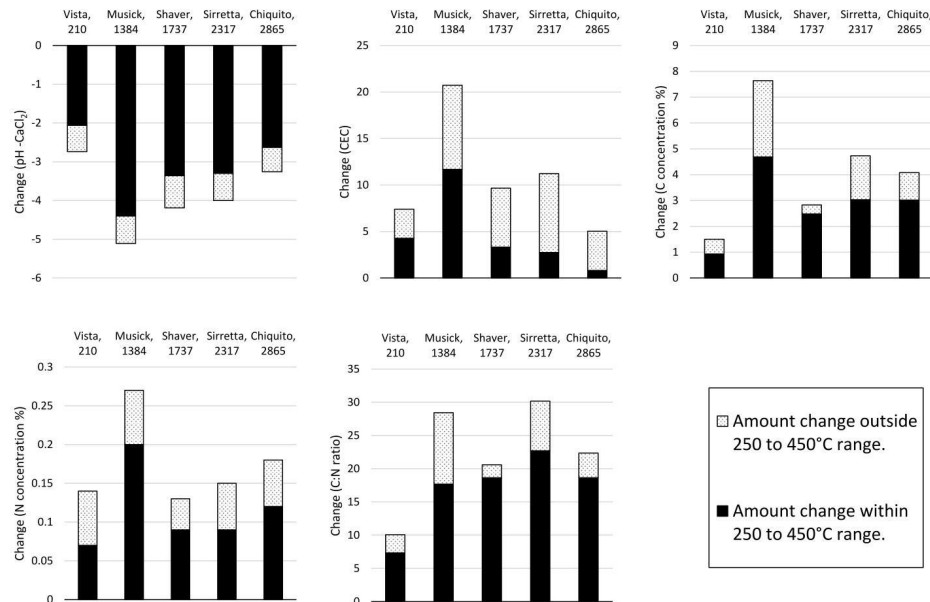

2   Figure 10: Total amount of change from unburned to 650 °C combusted soils showing amount

3   of change within the 250 to 450 °C range.