# Peer review of "Thermal alteration of soil physico-chemical properties: A"

_SOIL, 2016_

## Referee Comment (RC1) · Xinyan Huang (Referee) · 30 Mar 2016

This paper conducted very detailed and well-controlled experiments to study the influence of fire on top soils. The experimental results are clear and the discussion is reasonable. Some comments are provided to help improve the paper.

1. The use of word "combustion" should be avoided in the title and the main text. In general, it is possible to sustain a smoldering combustion in organic soils. However, it is questionable if such combustion is possible for high-mineral soils tested in this paper. In the furnace below 400C, the mass loss in soil is a mainly a result of pyrolysis

which produces pyrolysates and black chars which does not require oxygen, so it is not a combustion. When the furnace temperature exceeds 400C, chars are further oxidized which can be called as combustion. Without such high-temperature furnace, combustion may be sustain in soil. Therefore, using "combustion" and "combustion temperature" here can be misleading, instead "heating" or "soil or environmental temperature"

2. Both the fire heating temperature and heating duration determine the fire severity. In real fire, the duration for soil sustained in a higher temperature is usually shorter, rather than a fixed 30 min. Of course, in lab experiment controlling the heating duration makes a better comparison. But it is better to emphasize what is the real fire condition to avoid confusion.

3. The air supply during the heating is not mentioned in the paper. Is the air supply sufficient, or is the furnace sealed? The oxygen supply can significantly change the decomposition process of SOM.

4. In the paper, SOM is used very often, however, its value is not given for any soil samples. SOM should be easily measured, for example, by quantifying its inorganic matter after a complete oxidation in high-temperature oven. Comparatively, the organic carbon in soil is not so simple to quantify. Therefore, using SOM to correlate other parameters such as pH, CEC is more useful and reproduce current experiments with different soils. In fact, SOM correlates with C very well: generally increasing with the organic carbon. Correlating SOM will not alter the conclusions in this paper.

5. I recommend to split the discussion section and add a short discussion in each subsection of results. Most of experimental results are expected, and can be explained by a simple analysis right after showing the figure. It will also make readers easier to follow the discussion.

---

## Referee Comment (RC2) · Anonymous Referee #2 · 21 Apr 2016

This paper represents a detailed and thorough description of the physico-chemical properties of different experimentally burned soils taken from across a climosequence transect in Sierra Nevada. This study clearly represents a substantial body of work and such detailed observations will be of interest to the readership of SOIL. However, while I appreciate the amount of work that has gone into this study, I do have some concerns regarding (some of) the methodology and structuring of this paper (see comments below).

General comments

Title and Introduction

The title does not well reflect the content of this paper. It does not seem appropriate to refer to this study as a space-for-time substitution study as there is no discussion of the vegetation or how it may change in the future (no context for the soils). Nor is there sufficient discussion for how the climate is projected to change fire regimes specifically in the study area. Surely it would be more valuable instead to have the words "climosequence" and "Sierra Nevada" in the title?

The introduction could benefit from being expanded to include the current vegetation (and soils) and current fire regimes (intensity, severity, frequency etc.) specifically of the Sierra Nevada study area, and how these may change with anticipated climate warming. This additional information would then provide better context and rationale for this study.

Use of term fire intensity

Page 1, line 22 (and throughout): Suggest the term "fire intensity" is removed throughout the paper. Fire intensity (as defined by Keeley, 2009; Int. J. Wildfire; v.18) is an energy flux, and has been shown to be only weakly correlated with maximum temperatures or heating duration. The intensity classes in this paper have been based on maximum surface temperatures reached in various wildfires (page 5, line 25) yet there is no discussion of the duration these temperatures were held at, and whether (or not) this compares to the 30 minute heating duration used in the muffle furnace experiments. Also, wildfires cannot be represented by a singular temperature (i.e. the muffle furnace) as the temperature varies widely both spatially and temporally (see Alexander 1982, Can. J. Bot. v.60; Finney et al., 2015, PNAS). It is therefore more appropriate to refer to furnace temperatures alone, not intensity, throughout this paper.

Materials and Methods

Please clarify how soil sieved at 2mm can represent actual topsoil in the field. Also,

does this sieving and drying process change any of the intrinsic soil properties?

Use of a muffle furnace for combustion experiments As stated in the other referee comments, a muffle furnace does not fully capture the combustion process (only pyrolysis) that occurs during a wildfire and therefore the charcoals that are produced using this method should not be used to describe fire intensities (see comment above).

Discussion

Given the above comments on the use of charcoals produced in a muffle furnace to describe fire intensity, any subsequent interpretations of fire intensity should therefore be re-evaluated. The discussion section "4.2 climate change implications" could benefit from being expanded to discuss whether (and how) both the vegetation, and therefore soil, is expected to change in Sierra Nevada in the future in response to warming. If the rain-snow transition zone will move to higher altitudes, then will the treeline/ ecosystem boundaries in this climosequence also shift upwards? Vegetation is an important part of soil formation, yet other than in the study site and soil description section of the methods there is little to no discussion of the vegetation in this study area. This needs to be included in the text as fire behaviour is, in part, dependent on vegetation, so projected changes in vegetation first need to be discussed in order to comment on how the fire regimes in the study area may be altered in the future.

Specific comments

Check throughout: As several statistical tests have been used in this study where p values are given, or the term "significant" is used in the text, this should be followed by the statistical test used and the p value in brackets.

Page 2, line 3: what is the evidence for prescribed fires having temperatures <250°C? Is this soil temperature? Please provide a reference.

Page 2, line 5-6: there is no discussion of climate change scenarios or how fire intensity in this study area is anticipated to change in the future. Please remove this sentence

or expand the discussion.

Page 2, line 25: insert reference for fires "maintaining the health of ecosystems"

Page 4, line 1-3: suggest the study aims are rephrased as it is not possible to scale up muffle furnace experiments to the field, let alone to predict effects of different fire intensities on soil.

Page 6, line 24: please explain what a can is

Page 6, line 27-28: please rephrase this sentence. It is difficult to understand.

Page 7, line 7: please explain what a seven point measurement is.

Page 10, line 18-19: please clarify what is meant by kaolinite experiences loss at >550°C.

Page 12, line 30: add the temperature ranges of SOM combustion

Page 14, line 7: Please give a temperature range for "higher temperatures".

Page 15, line 23 and page 18, line 4-8: misuse of term "fire severity"

Page 16, line 6: please insert a reference for prescribed fires not contributing to soil C loss.

Page 18, line 18-23: suggest this paragraph is either re-phrased or removed. The rest of the paper refers to "fire intensity". There is no discussion of fire severity in this paper, or how different fire severities affect the physical properties of soil.

Technical and Typographical corrections

Page 1, Line 25: suggest change to "with increasing temperature"

Page 2, line 2: suggest rephrase to "occurred between 350°C and 450°C" to be more concise.

Page 2, line 15: wildland fire (one word)

Page 2, line 18: please change "significant changes on global..." to "significant changes to global..."

Page 2, line 21-22: suggest re-phrase, this sentence is difficult to understand. Perhaps something like "...climate change are likely to affect topsoils in fire-prone ecosystems"?

Page 3, line 1-2: is this information relevant to this study?

Page 3, line 4: insert "the" (depends on the fire intensity)

Page 3, line 7: suggest change "of" to "such as"

Page 4, line 27-29: suggest shorten this sentence for clarity to, "...developed under similar granitic parent material and are located in landscapes of similar age, relief, slope and aspect (Trumbore et al., 1996) with significant..."

Page 6, line 21: The methods section needs to be written in past tense (please check throughout). Please change "four grams of soil is weighed into sieve" to "four grams of soil was weighed into a sieve".

Page 6, line 22: please add the "the sample"

Page 6, line 23: please change to "Any soil passing through the sieves"

Page 7, line 8: replace "from" with "using"

Page 7, lines 11-12: add "a" "a fine powder", "a ball-mill" (please check throughout the paper. There are many instances of this).

Page 7, line 15: replace "at step interval" with "with a step interval"

Page 7, line 20: replace "with" with ","

Page 8, line 3: suggest change to "sieved <2mm soil samples were ground to a powder consistency using a ball-mill"

Page 8, line 5: how were the C and N values corrected?

Page 8, line 16-17: suggest delete "with increase in heating temperature, all the soils exhibited a similar trend in color change" as this is repetition from the preceding sentence.

Page 8, line 18: delete "with" and change "in mid" to "at mid"

Page 10, line 20: suggest re-phrase perhaps to "Gibbsite was also not found in soils heated to >450°C"

Page 10, line 29: change to "soils. Yet, all soils became alkaline" for clarity.

Page 11, line 15: "ranged". Also, could "less than 2%" and "over 7%" be more specific?

Page 11, line 19: Typo (p<0.05) described in text as insignificant.

Page 11, line 19,20: "The C:N ratio", "their C:N ratios"

Page 11, line 21: perhaps simplify to: "in a similar pattern to the C concentration"

Page 11, line 24: "the loss"

Page 12, line 4: "The topsoil"

Page 12, line 25: the link of reddening soils and weathering is irrelevant for this study. Suggest rephrase to "is likely a result of oxidation and the transformation of iron oxides that occurs during combustion"

Page 12, line 28: "topsoil"

Page 13, line 1: suggest add period after soils and add "For example, Musick (1384m) soils, had the..."

Page 13, line 4-6: suggest re-phrase the sentence beginning "the influence..." as this is difficult to understand.

Page 13, line 13: "the main"

Page 13, line 22: "an increase"

Page 13, line 24: "the aggregate"

Page 14, line 4: "an increase"

Page 14, line 5: "the removal"

Page 14, line 6: "the overall size"

Page 14, line 24: replace "to" with "with"

Page 14, line 27: "the start"

Page 14, line 28: "to this pH increase"

Page 18, line 17: suggest change "collapse of" to "disappearance of"

---

## Referee Comment (RC3) · G. M. Davies (Referee) · 3 May 2016

Overall I thought this paper was a useful contribution to our understanding of how variation in fire severity induces changes in soils across heterogeneous landscapes. The manuscript was well-structured and easy to follow and very clearly written. The lab methods were generally presented in detail and evaluated a wide range of metrics potentially influenced by fire. The figures and tables were generally very clear and nicely drawn. There was a small number of typos throughout and I've drawn attention to these in the annotated manuscript provided below. I think the paper is definitely suitable for publication but some major changes are required:

1) I have to admit that I wasn't particularly convinced by the climate change "story" the paper currently seeks to build itself on. First I didn't see how the design can really be considered a space for time substitution (indeed that's never really justified in detail in the introduction or methods). My guess was the authors are suggesting that the lower elevation sites are meant to emulate higher future temperatures. I would argue that's a pretty broad simplification and I'm not sure I can go with it - future higher temperatures will be super-imposed upon existing soil types (changing them) creating novel edaphic-climate combinations and new ecosite types. The present study is more of a sensitivity analysis examining differences in the response of soils to varying (simulated) fire intensity. That's certainly not uninteresting in itself and should be more than enough justification.

I think this is the strongest link the authors have to argue climate change implications of their work can be found right towards the end of the discussion (page 18). I'm not convinced about the climate series idea - soils are likely to change only slowly to climate change with very significant lag. The soils represent the results of underlying geological conditions and millenia of differing biological activity - surely that won't be erased overnight by climate change alone? What might happen is that alteration to disturbance regimes will alter vegetation and microbial communities and, in the process, alter soil properties and soil forming processes. I would urge the authors to recast their paper on more reasonable grounds

2) I thought the lab methods could do with greater justification and a greater consideration needs to be given as to whether the methods really emulate what happens during a wildfire in any useful way. For instance what might the implications of working with dried soils be? How does the presence of water in the soil affect physical and chemical processes during the passage of the fire front - why not study how moisture content and heating temperature affect changes? I would also like to see more thought about whether the heating times are appropriate. I would have preferred to see some study of the effects of heating duration. 30-40 minutes is a long time for a fire to be resident at a site. I warrant that it might approximate conditions under a smouldering log but then to what extent are you actually simulating changes more generally associated with a fire - logs occupy a small proportion of the soil surface.

3) There is room for improvement of the statistical analysis. Specifically: - Mixed models would be more appropriate (no need to average cores) - Stats (main test at least) need to be reported in full (even if in tables in supplementary material) - Data appears ripe for analysis with multivariate methods. Constrained ordination (e.g. Redundancy Analysis) would be particularly interesting as would allow you to test how changes in properties occur across temperatures and sites - have the authors considered such approaches? - New analyses are introduced in the Discussion section that were never described in the Methods, some of the results appear incomplete - the authors refer to doing regression analyses but only present a table of correlation coefficients.

The Discussion section generally very good with nice links made to previous similar or related studies. There did appears to be some confusion between the concepts of fire intensity and fire severity though. I recommend following the usage defined by Keeley which has been widely adopted: http://www.fs.fed.us/postfirevegcondition/documents/publications/keeley_ijwf_2009.pdf

Specific comments, corrections and requests for clarification can be found in the annotated manuscript attached.

Please also note the supplement to this comment:
http://www.soil-discuss.net/soil-2016-4/soil-2016-4-RC3-supplement.pdf

**Supplement:**

[revised manuscript text omitted]

*I like what the study is trying to do but at this point am rather skeptical of the climate angle. It seems to be more of a study of the effects of variation in simulated fire severity (~duration of heating of soil). Fire severity may change with a changing climate but also varies hugely between fires - the inference might be how varying fire severity (~ fire weather and fuel structure) would effect soils but it's something of a leap to go from this to climate change as there are likely to be complex interacting processes involved. To infer a climate effect you're at least two steps removed from the process you're studying (climate ~ fire weather ~ fire severity/behavior ~ soil chemico/physico/biological changes*

[revised manuscript text omitted]

*Margin note:* I think this is the strongest link you have to argue climate change implications. I'm not convinced about the climate series idea - soils are likely to change only slowly to climate change with very significant lag. The soils represent the results of underlying geological conditions and millenia of differing biological activity - surely that won't be erased overnight by climate change alone. What might happen is that alteration to disturbance regimes will alter vegetation and microbial communities and, in the process, alter soil properties and soil forming processes

[revised manuscript text omitted]

Perhaps highlight areas of the figure where you believe you're seeing important/interesting changes in peaks?

Figure 6: XRD diagram for Music series soils.

[Figure]

[Figure]

Figure 7: Relative amounts of minerals identified from powder XRD.

[Figure]

[Figure]

Figure 8: pH (geometric means) and cation exchange capacity (adjusted for mass loss) changes with increase in heating temperature. CEC values below the 2 cmol$_c$/kg are assigned a value of

1 for plotting. Error bars represent standard error where n=3. Different letters represent significantly different means (p<0.05) at each temperature after Tukey's HSD testing.

[Figure]

Figure 9: Carbon concentration, Nitrogen concentration and C:N atomic ratio changes with increase in heating temperature. Error bars represent standard error where n=3. Different letters represent significantly different means (p<0.05) at each temperature after Tukey's HSD testing.

[Figure]

[Figure]

Figure 10: Total amount of change from unburned to 650 °C combusted soils showing amount of change within the 250 to 450 °C range.

---

## Author Comment (AC1) · 21 May 2016

*Comments on the manuscript are followed by our responses. Text locations in the manuscript are indicated by a combination of page number and line number (page#:line#).*

**Comment 1.** The use of word "combustion" should be avoided in the title and the main text. In general, it is possible to sustain a smoldering combustion in organic soils. However, it is questionable if such combustion is possible for high-mineral soils tested

in this paper. In the furnace below 400C, the mass loss in soil is a mainly a result of pyrolysis which produces pyrolysates and black chars which does not require oxygen, so it is not a combustion. When the furnace temperature exceeds 400C, chars are further oxidized which can be called as combustion. Without such high-temperature furnace, combustion may be sustain in soil. Therefore, using "combustion" and "combustion temperature" here can be misleading, instead "heating" or "soil or environmental temperature"

**Author response:** We agree with this comment. Relevant occurrences of the word "combustion" in the manuscript were either removed (2:4; 3:22), or the sentences rewritten with appropriate terminology (3:19; 5:11; 11:24; 12:6-8; 14:3-6, 12; 15:27,30; 16:25-27; 17:4; 18:15-16).

**Comment 2.** Both the fire heating temperature and heating duration determine the fire severity. In real fire, the duration for soil sustained in a higher temperature is usually shorter, rather than a fixed 30 min. Of course, in lab experiment controlling the heating duration makes a better comparison. But it is better to emphasize what is the real fire condition to avoid confusion.

**Author response:** Justifications for the heating duration used was given in section 2.2 (5: 29 - 6: 10). We have decided to expand the discussion to highlight heating duration in relation to our methods in the revised manuscript. We have also rewritten a paragraph in introduction (3:14-18) to clarify the importance of heating duration in fires.

**Comment 3.** The air supply during the heating is not mentioned in the paper. Is the air supply sufficient, or is the furnace sealed? The oxygen supply can significantly change the decomposition process of SOM.

**Author response:** A sentence explaining the oxygen supply is added in methods section (5:21). All soil heating procedure was done by Thermo Scientific Thermolyne

Largest Tabletop Muffle Furnace (Thermo Fisher Scientific Inc., 81 Wyman Street Waltham, MA USA 02451). The furnace air supply was not considered limiting for the following reasons: The furnace was not sealed, and the furnace had an internal capacity of 45 L and the volume of soil in the furnace at a time was approximately 0.924 L (i.e. volume of crucible multiplied by 24 crucibles per run: $(\pi 3.5^2 \times 1) \times 24 = 924 cm^3$

**Comment 4.** In the paper, SOM is used very often, however, its value is not given for any soil samples. SOM should be easily measured, for example, by quantifying its inorganic matter after a complete oxidation in high-temperature oven. Comparatively, the organic carbon in soil is not so simple to quantify. Therefore, using SOM to correlate other parameters such as pH, CEC is more useful and reproduce current experiments with different soils. In fact, SOM correlates with C very well: generally increasing with the organic carbon. Correlating SOM will not alter the conclusions in this paper.

**Author response:** Our usage of SOM in our findings is as a general descriptor for organic compounds in soil, however when a specific data is being discussed we have used the quantity of C which was actually quantified accurately.

**Comment 5.** I recommend to split the discussion section and add a short discussion in each subsection of results. Most of experimental results are expected, and can be explained by a simple analysis right after showing the figure. It will also make readers easier to follow the discussion.

**Author response:** We agree with this comment and we have made necessary changes with the discussion to address the reviewer's concern.

*We appreciate the thoughtful comments from the reviewer. Thank you!*

---

## Author Comment (AC2) · 21 May 2016

*Comments on the manuscript are followed by our responses.   Text locations in the manuscript are indicated by a combination of page number and line number (page#:line#).*

**General Comments**

**Comment 1.** Title: The title does not well reflect the content of this paper. It does not seem appropriate to refer to this study as a space-for-time substitution study as there is no discussion of the vegetation or how it may change in the future (no context for

the soils). Nor is there sufficient discussion for how the climate is projected to change fire regimes specifically in the study area. Surely it would be more valuable instead to have the words "climosequence" and "Sierra Nevada" in the title?

**Author response:** We agree with this comment and we have revised the title to better reflect the experiment as, "Thermal alteration of soil physico-chemical properties: A systematic study to infer response of Sierra Nevada climosequence soils to forest fires."

**Comment 2.** Introduction: The introduction could benefit from being expanded to include the current vegetation (and soils) and current fire regimes (intensity, severity, frequency etc.) specifically of the Sierra Nevada study area, and how these may change with anticipated climate warming. This additional information would then provide better context and rationale for this study.

**Author response:** We have expanded the discussion on vegetation and fire regimes of study site by adding a paragraph in section 2.1.

**Comment 3.** Use of term fire intensity: Page 1, line 22 (and throughout): Suggest the term "fire intensity" is removed throughout the paper. Fire intensity (as defined by Keeley, 2009; Int. J. Wildfire; v.18) is an energy flux, and has been shown to be only weakly correlated with maximum temperatures or heating duration. The intensity classes in this paper have been based on maximum surface temperatures reached in various wildfires (page 5, line 25) yet there is no discussion of the duration these temperatures were held at, and whether (or not) this compares to the 30 minutes heating duration used in the muffle furnace experiments. Also, wildfires cannot be represented by a singular temperature (i.e. the muffle furnace) as the temperature varies widely both spatially and temporally (see Alexander 1982, Can. J. Bot. v.60; Finney et al., 2015, PNAS). It is therefore more appropriate to refer to furnace temperatures alone, not intensity, throughout this paper.

**Author response:** We agree with this comment. While maximum surface temperature is often used as a metric of fire intensity, especially for studies of direct fire effect on soil (e.g. Neary D.G. et al. (1999) Forest Ecology and Management), it is more accurate to directly refer to the temperature. We have clarified statements where maximum temperature is used to loosely infer fire intensity.

**Comment 4.** Materials and Methods: Please clarify how soil sieved at 2mm can represent actual topsoil in the field. Also, does this sieving and drying process change any of the intrinsic soil properties?

**Author response:** We have followed these steps to essentially extract soils in the strict definition, and to remove large rock fragments and undecomposed organic matter. The sieving and oven drying processes do alter soil some soil properties but we think these alterations have minimal effect on the soil properties we measured. The measurement procedures for these properties themselves often introduce more alterations. The significant effect of drying has been discussed and we used dry samples to control for moisture effects that are known to cause aggregate stability changes and we have acknowledged this in the discussion of aggregate stability (5:16-21).

**Comment 5.** Use of a muffle furnace for combustion experiments As stated in the other referee comments, a muffle furnace does not fully capture the combustion process (only pyrolysis) that occurs during a wildfire and therefore the charcoals that are produced using this method should not be used to describe fire intensities (see comment above).

**Author response:** The pyrogenic products that would be produced from this experiment are necessarily derived from soil organic matter. We have avoided the use of the term fire intensity except where we use 'intensity' in the general term as maximum temperature is a matric of intensity.

**Comment 6.** Discussion: Given the above comments on the use of charcoals pro-
duced in a muffle furnace to describe fire intensity, any subsequent interpretations of
fire intensity should therefore be re-evaluated. The discussion section "4.2 climate
change implications" could benefit from being expanded to discuss whether (and how)
both the vegetation, and therefore soil, is expected to change in Sierra Nevada in the
future in response to warming. If the rain-snow transition zone will move to higher al-
titudes, then will the treeline/ ecosystem boundaries in this climosequence also shift
upwards? Vegetation is an important part of soil formation, yet other than in the study
site and soil description section of the methods there is little to no discussion of the
vegetation in this study area. This needs to be included in the text as fire behaviour
is, in part, dependent on vegetation, so projected changes in vegetation first need to
be discussed in order to comment on how the fire regimes in the study area may be
altered in the future.

**Author response:** We have revised the title and the objectives of the paper to clarify
our objectives as a study focused on first order effect of heating on soils. Hence,
we expect our findings to provide primary information that should be of vital interest
in detailed fire and ecological studies. However, given the limited level of depth our
experiment goes into exploring fire in the ecosystem we believe the level of detail in
discussion section on Sierra Nevada ecology is sufficient.

**Specific Comments**

Throughout: As several statistical tests have been used in this study where p values
are given, or the term "significant" is used in the text, this should be followed by the
statistical test used and the p value in brackets.

**Author response:** Because the statistical tests in this study are the same type, we
had decided to mention the statistical analysis methods in the methods section to avoid
repetition. However, to address this comment, we have added details of the statistical
analysis and placed the statistical analysis in its own subsection (as Section 2.4).

Page 2, line 3: what is the evidence for prescribed fires having temperatures <250 °C? Is this soil temperature? Please provide a reference.

**Author response:** We have removed the reference to prescribed fires. While fire intensity of prescribed fires is typically kept low, a 250 °C cutoff may not be appropriate for forest ecosystems (for example: Busse et al. (2005) Int. J. of Wildland Fire; Garcia-Corona et al. (1999.) Geoderma).

Page 2, line 5-6: there is no discussion of climate change scenarios or how fire intensity in this study area is anticipated to change in the future. Please remove this sentence or expand the discussion.

**Author response:** We have rewritten the section to clarify our objectives. Page 2, line

25: insert reference for fires "maintaining the health of ecosystems"

**Author response:** We have added a reference.

Page 4, line 1-3: suggest the study aims are rephrased as it is not possible to scale up muffle furnace experiments to the field, let alone to predict effects of different fire intensities on soil.

**Author response:** Given that the impact of fire on soil is primarily by the input of heat (which is dependent on temperature and also the duration), we believe such muffle furnace simulations can provide valuable data that can be useful to infer the effect of fires. We believe that such research and our findings provide valuable information on the first order effects of heat input into soils which provides at least a baseline information of how soils respond to different temperature. In addition to giving insight into different fires, our findings can also give insights on soil response across the vertical temperature gradient that occur with vegetation fires.

Page 6, line 24: please explain what a can is

[Figure]

**Author response:** The cans are part of the wet-sieving apparatus into which the sieve is immersed for wet-sieving. As they have no unique use in the procedure, the paragraph has been rewritten without mention of the cans.

Page 6, line 27-28: please rephrase this sentence. It is difficult to understand.

**Author response:** The paragraph has been rewritten to clarify the procedure.

Page 7, line 7: please explain what a seven point measurement is.

**Author response:** The sentence has been rewritten to indicate that seven measurement points are taken to calculate specific surface are using the N2-BET isotherm. The seven-point measurement is an accuracy configuration within the instrument software setup which internally calculates the specific surface area.

Page 10, line 18-19: please clarify what is meant by kaolinite experiences loss at >550 C.

**Author response:** The sentence is rewritten to clarify that from all the minerals, kaolinite concentration showed the largest decrease and that decrease happened at temperatures above 550 ° C.

Page 12, line 30: add the temperature ranges of SOM combustion

**Author response:** The temperature ranges where SOM combusts has been added.

Page 14, line 7: Please give a temperature range for "higher temperatures".

**Author response:** We have indicated a temperature above which thermal alterations were noted.

Page 15, line 23 and page 18, line 4-8: misuse of term "fire severity"

**Author response:** We have removed the reference to fire severity.

Page 16, line 6: please insert a reference for prescribed fires not contributing to soil C loss.

**Author response:** We have rewritten the preceding sentence to clarify that we are discussing the implication of our findings. We are not citing from literature.

Page 18, line 18-23: suggest this paragraph is either re-phrased or removed. The rest of the paper refers to "fire intensity". There is no discussion of fire severity in this paper, or how different fire severities affect the physical properties of soil.

**Author response:** The paragraph has been rewritten for clarity.

**Technical and Typographical corrections**

We have accepted all technical and typographical comments given.

*We appreciate the thoughtful comments from the reviewer. Thank you!*

---

## Author Comment (AC3) · 21 May 2016

*Comments on the manuscript are followed by our responses. Text locations in the manuscript are indicated by a combination of page number and line number (page#:line#).*

**General Comments**

**Comment 1.** I have to admit that I wasn't particularly convinced by the climate change "story" the paper currently seeks to build itself on. First I didn't see how the design can really be considered a space for time substitution (indeed that's never really justified

in detail in the introduction or methods). My guess was the authors are suggesting that the lower elevation sites are meant to emulate higher future temperatures. I would argue that's a pretty broad simplification and I'm not sure I can go with it - future higher temperatures will be super-imposed upon existing soil types (changing them) creating novel edaphic-climate combinations and new ecosite types. The present study is more of a sensitivity analysis examining differences in the response of soils to varying (simulated) fire intensity. That's certainly not uninteresting in itself and should be more than enough justification.

I think this is the strongest link the authors have to argue climate change implications of their work can be found right towards the end of the discussion (page 18). I'm not convinced about the climate series idea - soils are likely to change only slowly to climate change with very significant lag. The soils represent the results of underlying geological conditions and millenia of differing biological activity - surely that won't be erased overnight by climate change alone? What might happen is that alteration to disturbance regimes will alter vegetation and microbial communities and, in the process, alter soil properties and soil forming processes. I would urge the authors to recast their paper on more reasonable grounds

**Author response:** We accept the reviewer's thoughtful comments. We have revised the title and other parts of the manuscript to address these concerns. We have revised the objectives of this study as to further our understanding of the effect of different heating temperatures on soil. We have limited our inferences to climate change induced fire regime change only to the discussion section where general conclusions can be drawn from the study.

**Comment 2.** I thought the lab methods could do with greater justification and a greater consideration needs to be given as to whether the methods really emulate what happens during a wildfire in any useful way. For instance what might the implications of working with dried soils be? How does the presence of water in the soil affect physical and chemical processes during the passage of the fire front - why not study how moisture content and heating temperature affect changes? I would also like to see more thought about whether the heating times are appropriate. I would have preferred to see some study of the effects of heating duration. 30-40 minutes is a long time for a fire to be resident at a site. I warrant that it might approximate conditions under a smouldering log but then to what extent are you actually simulating changes more generally associated with a fire - logs occupy a small proportion of the soil surface.

**Author response:** Justification for using dry soils was to avoid heating rate effects on moist soils (at temperatures near 100 °C). Since all treatment temperatures were above boiling point, the oven drying ensured there was minimal heating rate effect on soils. The justifications for the heating duration used was given in section 2.2 (5: 29 - 6: 10). We have decided to expand the discussion to highlight heating duration in relation to our methods in the revised manuscript. We have also rewritten a paragraph in introduction (3:14-18) to clarify the importance of heating duration in fires.

**Comment 3.** There is room for improvement of the statistical analysis. Specifically: - Mixed models would be more appropriate (no need to average cores) - Stats (main test at least) need to be reported in full (even if in tables in supplementary material) - Data appears ripe for analysis with multivariate methods. Constrained ordination (e.g. Redundancy Analysis) would be particularly interesting as would allow you to test how changes in properties occur across temperatures and sites - have the authors considered such approaches? - New analyses are introduced in the Discussion section that were never described in the Methods, some of the results appear incomplete - the authors refer to doing regression analyses but only present a table of correlation coefficients.

**Author response:** There is a potential of using mixed models, and redundancy analysis type tests. However, for the purposes of this study, we feel using mean with analysis of variance gives sufficient results. We agree with reporting the statistical analysis in

full. We have included the ANOVA tables and Tukey's HSD comparison of means statistics to be presented as supplementary material. An explanation of the methods used in the regression analysis introduced in the discussion is also included in the methods section.

**Comment 4.** The Discussion section generally very good with nice links made to previous similar or related studies. There did appears to be some confusion between the concepts of fire intensity and fire severity though. I recommend following the usage defined by Keeley (2009) which has been widely adopted.

**Author response:** We agree with the comment. We have fixed all references to intensity and severity to be consistent with that defined by Keeley (2009).

**Specific Comments (Comments from supplement)**

We have addressed all comments and corrections related to the writing and grammar.

*We appreciate the thoughtful comments from the reviewer. Thank you!*

---

## Author Response (AR1)

Dear Antonio Jordan (Topical Editor),

Following is the revised version of manuscript with track-changes mark-up. Changes have been highlighted using three different colors based on reviewer comments. Changes based on first reviewer comments are highlighted yellow; changes based on second reviewer comments are highlighted green; and changes based on third reviewer comments are highlighted blue.

Sincerely,

Samuel Araya (on behalf of all authors)

[revised manuscript text omitted]

---

## Author Response (AR2)

Dear Antonio Jordan (Topical Editor),

Following is the revised version of manuscript with track-changes mark-up. We have included all the comments.

We appreciate the thoughtful comments.

Thank you,

Samuel Araya (on behalf of all authors)

[revised manuscript text omitted]